# Optimization Inspired Multi-Branch Equilibrium Models

**Mingjie Li[1], Yisen Wang[1,2], Xingyu Xie[1], Zhouchen Lin[1,2,3]** *

[1] Key Lab. of Machine Perception (MoE), School of Artificial Intelligence, Peking University
[2] Institute for Artificial Intelligence, Peking University
[3] Pazhou Lab, Guangzhou, 510330, China.

## Abstract

Works have shown the strong connections between some implicit models and optimization problems. However, explorations on such relationships are limited. Most works pay attention to some common mathematical properties, such as sparsity. In this work, we propose a new type of implicit model inspired by the designing of the systems' hidden objective functions, called the Multi-branch Optimization induced Equilibrium networks (MOptEqs). The model architecture is designed based on modelling the hidden objective function for the multi-resolution recognition task. Furthermore, we also propose a new strategy inspired by our understandings of the hidden objective function. In this manner, the proposed model can better utilize the hierarchical patterns for recognition tasks and retain the abilities for interpreting the whole structure as trying to obtain the minima of the problem's goal. Comparing with the state-of-the-art models, our MOptEqs not only enjoys better explainability but are also superior to MDEQ with less parameter consumption and better performance on practical tasks. Furthermore, we also implement various experiments to demonstrate the effectiveness of our new methods and explore the applicability of the model's hidden objective function.

## 1 Introduction

Whereas Deep Neural Networks (DNNs) have achieved great success in many real-world tasks such as computer vision and neural language process, the limited interpretability of DNNs greatly hinders their further development.

However, many traditional machine learning methods, e.g.,matrix recovery (Zhang et al., 2018b; 2015; Liu & Li, 2016), subspace clustering (You et al., 2016), image deblurring (Liu et al., 2014) and so on, can be interpreted into minimizing the following objective functions:

$$\min_{\mathbf{Z},\mathbf{E}} f(\mathbf{Z}) + g(\mathbf{E}), \ s.t. \ \mathbf{X} = \mathbf{A}\mathbf{Z} + \mathbf{B}\mathbf{E}.$$

where $\mathbf{A} \in \mathbb{R}^{m \times d_1}$, $\mathbf{B} \in \mathbb{R}^{m \times d_2}$, $\mathbf{X} \in \mathbb{R}^{m \times n}$, and $f(\cdot)$ and $g(\cdot)$ are convex functions designed by the modelling of their properties. For this count, the interpretability of such methods is much better than DNNs. Furthermore, these methods can also enjoy state-of-the-art and robust performance on these tasks. We call such interpretability these methods enjoy as "mathematical interpretability", i.e., whether the whole network structure can be summarized as a compact mathematical model that can be mathematically analyzed.

Except for these methods, the Optimization Induced Equilibrium Networks (OptEqs) proposed by Xie et al. (2021) recover its whole system to an optimization problem. The forward propagation for OptEqs tries to solve Eqn (1) to get output $\mathbf{y}, \mathbf{z}^* \in \mathbb{R}^d$ for input $\mathbf{x} \in \mathbb{R}^{\mathbf{d_{in}}}$. We call the front part of Eqn (1) as OptEqs' equilibrium equation, which is the central part for equilibrium models. And its forward procedure can be regarded as solving the hidden objective functions shown as Eqn(2),

---

*Corresponding author

$$\mathbf{z}^* = \sigma(\mathbf{W}\mathbf{z}^* + \mathbf{U}g(\mathbf{x}) + \mathbf{b}), \ \mathbf{y} = \mathbf{W}_c\mathbf{z}^*, \tag{1}$$

$$\min_{\mathbf{z}} G(\mathbf{z}; g(\mathbf{x})) = \min_{\mathbf{z}} \left[ \mathbf{1}^\top f(\mathbf{W}^{-1\top}\mathbf{z}) - \left\langle \mathbf{U}g(\mathbf{x}) + \mathbf{b}, \mathbf{W}^{-1\top}\mathbf{z} \right\rangle + \frac{1}{2}\|\mathbf{W}^{-1\top}\mathbf{z}\|^2 - \frac{1}{2}\|\mathbf{z}\|^2 \right]. \tag{2}$$

$\mathbf{W}, \mathbf{U}, \mathbf{W}_c$ are learnable weights, $g$ is the convolution layers projecting input $\mathbf{x}$ from real-world domain to feature space like other neural architectures, and $f$ is the outputs' penalty.

Although OptEqs' interpretability is satisfying, it still remains some weaknesses worth exploring. First, although OptEqs performs better than some implicit models, these models only consider a single view (or resolution) of the input in their implicit parts. We call these models single-view implicit models in the following. However, nearly all the state-of-the-art pattern recognition systems (Lee et al., 2009b; Wang et al., 2019a; Huang et al., 2017; He et al., 2016; Burt & Adelson, 1987) benefit from the multi-layer or multi-resolution feature extractors in domains like computer vision and audio processing. Secondly, limited new modules are proposed in their work, which mainly consider the math properties of features like sparsity, but whether these properties can benefit image recognition tasks is unexplored.

Besides OptEqs, MDEQ (Bai et al., 2020) constructs its equilibrium equation $\mathbf{z}^* = \mathcal{F}(\mathbf{z}^*; \mathbf{x})$ and block $\mathcal{F}$ with the inspiration of explicit models especially HRNet Wang et al. (2019a). Then they solve the equilibrium equation by the accelerated algorithm for output $\mathbf{z}^*$. Although it shows the state-of-the-art performance with efficient memory cost as other implicit models, such strategies will make the model lose "interpretability" since the MDEQ is a black box because its output is solved in an implicit way (root-finding method), which makes analysis on its features of intermediate layers almost impossible. Furthermore, its complicated structure also hinders its analysis from mathematical systems. More efficient blocks with good interpretability still needs exploring.

Motivated by the limitations of the above models, we would like to design a multi-scale DEQ structure with state-of-the-art performance and preserve mathematical interpretability from the inspiration of OptEqs and multi-scale models. Our contributions are listed below:

- We propose a multi-branch implicit model, called Multi-branch OptEqs (MOptEqs), which efficiently utilizes different scales inputs with better performance and smaller model size. Furthermore, it still retains its connection to an optimization problem.

- We propose some properties modelled as new terms for the model's hidden objective function from our analysis on the relationships between branches and their hierarchical dependencies. With new problem's formulation, we obtain the fusion module in our MOptEqs, called Hierarchical Heritage and Diversity Module (HH&D Module).

- Apart from the new module design, we also propose the Perturbation Enhanced strategy (PE) for our MOptEqs from our analysis on the model's hidden objective function. The new strategy cannot only enhance the performance of our MOptEqs, but also improve the robustness of our models.

## 1.1 Related Works

**Implicit Models.** Nearly all modern deep learning approaches use *explicit* models, which provide an explicit computation graph for the forward propagation. In contrast, the computation graphs for *implicit* models are "flexible" or can be assumed as having "infinite" depth. For example, Neural ODEs (Chen et al., 2018; Massaroli et al., 2020) encode their neural architectures by a differential system with learnable parameters. Then the implicit ODE solvers they used is equivalent to a continuous ResNet taking infinitesimal steps. Furthermore, the training process of Neural ODEs can be depicted as finding a differential system of a certain type (like heat equations) by updating its learnable parameters which demonstrate that Neural ODE also enjoy interpretability to a certain extent.

Moreover, DEQ (Bai et al., 2019; Winston & Kolter, 2020) is another class of implicit models. The central part of DEQ is the designation of the equilibrium equations $\mathbf{z}^* = \mathcal{F}(\mathbf{z}^*; \mathbf{x})$ and block $\mathcal{F}$. Its forward procedure is trying to solve the equilibrium equations with input $\mathbf{x}$ to

get the equilibrium state $\mathbf{z}^*$ as output by accelerated algorithms. Since the $\mathbf{z}^*$ is fixed given $\mathcal{F}$ and $\mathbf{x}$, its inference can be regarded as forwarding an explicit network stacked by the same block $\mathcal{F}$ for infinite times. For example, DEQ chooses their $\mathcal{F}$ by one set of Conv+ReLU while MDEQ constructs an HRNet-like block as theirs. However, no evidence shows that the best block constructions in explicit models can perform the best in the DEQ scheme. The construction of DEQ blocks or equilibrium equations is an open question worth exploring. Furthermore, most DEQ blocks do not have any mathematical insights (like the diffusion process in Neural ODE) and perform totally a black box with limited interpretability.

Since implicit models usually adopt accelerated root-finding algorithms for their forward outputs and backward gradients, they enjoy the advantages of constant and more efficient memory costs compared with DNNs. Due to the above advantages, the design of implicit models draws much attention these days (Ghaoui et al., 2019; Gould et al., 2019). Apart from the models illustrated above, many kinds of other implicit models are proposed, including differentiable physics engines (Qiao et al., 2020; de Avila Belbute-Peres et al., 2018), logical structure learning (Wang et al., 2019b) and implicit neural blocks (Li et al., 2020).

**Model Interpretaiblity.** Many researchers nowadays are trying to make their proposed method more interpretable. Although there are many kinds of approaches to achieve this goal, we marginally divide them into two parts: "Empirical" and "Mathematical" interpretability. Many works, such as (Zhang et al., 2018a; Zhang & Zhu, 2018; Bau et al., 2018), attempt to empirically disassemble the black box by characterizing some statistical or structural information like the outputs and gradients of neural networks' middle layers. However, these works cannot be directly implemented on DEQ models since implicit ones do not have explicit depths like DNNs, which makes analyzing such models by dissecting each layer's behavior almost impossible. Apart from that, researchers (Djolonga & Krause, 2017; Amos & Kolter, 2017; Xie et al., 2019; Chan et al., 2020) also managed to understand the neural architecture by linking it to a mathematical problem. In this way, researchers can analyze the black box by its corresponding mathematical problem. Furthermore, new components can be proposed due to the analysis of these problems. Our model aims to design a new DEQ architecture with proper mathematical interpretability. Compared with former works, our model can perform better on the classification tasks with novel components.

## 2 Multi-Branch Optimized Induced Equilirium Models

### 2.1 The proposed architecture for the Multi-Branch OptEqs

Inspired by OptEqs, we first design an objective function for our tasks to solve and then use its first-order stationary conditions as MOptEqs' equilibrium equation. The base of our function is formed by summarizing several objective functions of different scales defined for OptEqs (Details are stated in Appendix A.1). However, such a model can not obtain satisfying results since these branches are independent. For this count, we need to design some terms describing the dependencies of each branch in the hidden objective function, as shown in the following paragraphs. Then we can obtain the equilibrium equations for our MOptEqs and our MOptEqs' block $\mathcal{F}$ by analyzing the first-order stationary condition of our designed optimization problem.

**Hierarchical Heritage Modeling.** State-of-the-art explicit models for image tasks are explicitly structured into sequential stages and process different resolutions hierarchically (He et al., 2016; Shelhamer et al., 2017; Lee et al., 2009a), which implies that features should be extracted hierarchically. In other words, the posterior branches should inherit from their prior ones, which we call such property hierarchical heritage. In our work, we are going to formulate such correlation of neighbouring branches in the hidden objective function by adding the following inner-product term into its origin:

$$\mathcal{H}(\mathbf{z}_i, \mathbf{z}_j) = \mathbf{z}_i^\top \mathbf{P}_{i:j} \mathbf{z}_j \tag{3}$$

where $\mathbf{z}_i \in \mathbb{R}^{d_i}, \mathbf{z}_j \in \mathbb{R}^{d_j}, \mathbf{P}_{i:j} \in \mathbb{R}^{s_i \times s_j}$ denotes the Average Downsample (its transpose can be regarded as a weighted nearest upsample) or Identity matrix suited to the shape of $\mathbf{z}_i$ and $\mathbf{z}_j$ ($i < j$ or $j = 1, i = L$). This term estimates the summation for similarity of $i$-th and $j$-th branchs channels with the same channel index (*corresponding channels*).

When $\mathbf{z}_i$ and $\mathbf{z}_j$ are similar, the inner product will be large. Otherwise, the result will become small, which implies that the relationship of corresponding channels is weak. In this way, we can ensure the similarity of corresponding channels of the near branches. Except for the relations between different branches' corresponding channels, the relationships between dis-corresponding channels are explored in the following section.

**Diversity Modeling.** In addition to the hierarchical relations for corresponding channels, works (Pang et al., 2019; Amada et al., 2021) have shown enhancing the diversities between branches can also improve the model's improvements. Therefore, we also consider making different branches extract various features to improve the representation abilities. To achieve such a goal, we add a diversity term in the objective function Eqn (4) for $i < j$,

$$\mathcal{D}(\mathbf{z}_i, \mathbf{z}_j) = \sum_{k_1=1}^{C} \sum_{\substack{k_2=1 \\ k_2 \neq k_1}}^{C} |\mathbf{vec}^{-1}(\mathbf{z}_i)^{(k_1)\top} \mathbf{vec}^{-1}(\mathbf{P}_{i:j}\mathbf{z}_j)^{(k_2)}|, \tag{4}$$

where $\mathbf{vec}^{-1}$ is the inverse vectorization operator converting the vectorized feature $\mathbf{z}_i \in \mathbb{R}^{d_i \times 1}$ to a matrix in $\mathbb{R}^{\frac{d_i}{C} \times C}$ with $C$ channels and $\mathbf{vec}^{-1}(\mathbf{z}_i)^{(k)} \in \mathbb{R}^{\frac{d_i}{C} \times 1}$ denotes the $k$-th channel of $i$-th branch in practice. Since we have already considered the relationships between the corresponding channels in the hierarchical term, we only tries to estimate the diversities of dis-corresponding channels of the $i$-th and $j$-th branches in this term. As the diversity term goes small, the diversities between the branches become stronger.

**The architecture of MOptEqs and its hidden problem.** With the two terms we proposed, we can reformulate the hidden objective for our problem as follows:

$$\min_{\mathbf{z}_1,...,\mathbf{z}_L} G(\mathbf{z}_1,...,\mathbf{z}_L; g(\mathbf{x})) = \min_{\mathbf{z}_1,...,\mathbf{z}_L} \sum_{i=1}^{L} \left[ \mathbf{1}^\top f(\mathbf{W}_i^{-1\top}\mathbf{z}_i) - \left\langle \mathbf{U}_i g(\mathbf{x}) + \mathbf{b}_i, \mathbf{W}_i^{-1\top}\mathbf{z} \right\rangle \right.$$
$$\left. + \frac{1}{2}(\lambda \mathcal{D}(\mathbf{z}_i, \mathbf{z}_{i+1}) + \|\mathbf{W}^{-1\top}\mathbf{z}_i\|^2 - \mathcal{H}(\mathbf{z}_i, \mathbf{z}_{i+1})) \right]. \tag{5}$$

$g(\mathbf{x})$ is the input feature for the raw input $\mathbf{x}$, $\lambda > 0$ is a hyperparameter and $\mathbf{z}_i \in \mathbb{R}^{d_i \times 1}$ are the final outputs of MOptEqs. We can choose $f$ to constrain $\mathbf{z}_i$ on our demand and will influence model's activation function. If we choose $f(x) = \mathbb{I}\{x \geq 0\}$ to ensure the outputs to be positive, then the activation function is ReLU. We set $\mathcal{D}(\mathbf{z}_L, \mathbf{z}_{L+1}) = \mathcal{D}(\mathbf{z}_L, \mathbf{z}_1)$ and $\mathcal{H}(\mathbf{z}_L, \mathbf{z}_{L+1}) = \mathcal{H}(\mathbf{z}_L, \mathbf{z}_1)$ when $i = L$ to complete the loop. The two terms we added tries to make the corresponding channels of different branches correlated by maximizing the $\mathcal{H}$ term and enhancing the diversities of different branches by minimizing $\mathcal{D}$ term. We note that $\mathcal{H}$ and $\mathcal{D}$ are not conflict since they are handling different pairs of channels for branches.

As the following proposition shows, we finally get the equilibrium equations (Eqn (6)) for our MOptEqs by calculating the first-order stationary conditions $\nabla_{\mathbf{z}_i} G = 0$ for problem $G$.

**Proposition 1** *The proposed multi-branch structure induced by Eqn (5) can be depicted as solving the equilibrium points* $\widetilde{\mathbf{z}}^* := [\mathbf{z}_1^{\top*},...,\mathbf{z}_L^{\top*}]^\top \in \mathbb{R}^{\sum_{i=1}^{L} d_i}$ *for the following equations:*

$$\widetilde{\mathbf{z}} = \widetilde{\mathbf{W}}^\top \sigma(\widetilde{\mathbf{W}} h(\tilde{\mathbf{z}}) + \widetilde{\mathbf{U}} g(\mathbf{x}) + \widetilde{\mathbf{b}})$$
$$where \ \widetilde{\mathbf{W}} = \begin{bmatrix} \mathbf{W}_1 & & \\ & \ddots & \\ & & \mathbf{W}_L \end{bmatrix}, \widetilde{\mathbf{U}} = [\mathbf{U}_1^\top,...,\mathbf{U}_L^\top]^\top, \widetilde{\mathbf{b}} = [\mathbf{b}_1^\top,...,\mathbf{b}_L^\top]^\top. \tag{6}$$

*And* $h(\tilde{\mathbf{z}}) = [h_1(\mathbf{z}_1, \mathbf{z}_L, \mathbf{z}_2)^\top,.., h_i(\mathbf{z}_i, \mathbf{z}_{i-1}, \mathbf{z}_{i+1})^\top,.., h_L(\mathbf{z}_L, \mathbf{z}_{L-1}, \mathbf{z}_1)^\top]^\top$ *and each* $h_i$ *is defined as the mapping from* $\mathbb{R}^{d_i} \times \mathbb{R}^{d_{i+1}} \times \mathbb{R}^{d_{i-1}}$ *to* $\mathbb{R}^{d_i}$,

$$h(\mathbf{z}_i, \mathbf{z}_{i-1}, \mathbf{z}_{i+1}) = \frac{1}{2}(\mathbf{P}_{i:i+1}\mathbf{z}_{i+1} + \mathbf{P}_{i-1:i}^\top\mathbf{z}_{i-1})$$
$$- \frac{\lambda}{2}\mathbf{vec}(\mathbf{vec}^{-1}(\mathbf{P}_{i:i+1}\mathbf{z}_{i+1})\mathbf{sign}[\mathbf{M}_{i:i+1} - \mathbf{diag}(\mathbf{M}_{i:i+1})])$$
$$- \frac{\lambda}{2}\mathbf{vec}(\mathbf{vec}^{-1}(\mathbf{P}_{i-1:i}^\top\mathbf{z}_{i-1})\mathbf{sign}[\mathbf{M}_{i-1:i}^\top - \mathbf{diag}(\mathbf{M}_{i-1:i})]),$$
$$\mathbf{M}_{i:i+1} = \mathbf{vec}^{-1}(\mathbf{P}_{i:i+1}\mathbf{z}_{i+1})^\top\mathbf{vec}^{-1}(\mathbf{z}_i),$$
$$\mathbf{M}_{i-1:i} = \mathbf{vec}^{-1}(\mathbf{z}_i)^\top\mathbf{vec}^{-1}(\mathbf{P}_{i-1:i}^\top\mathbf{z}_{i-1}).$$

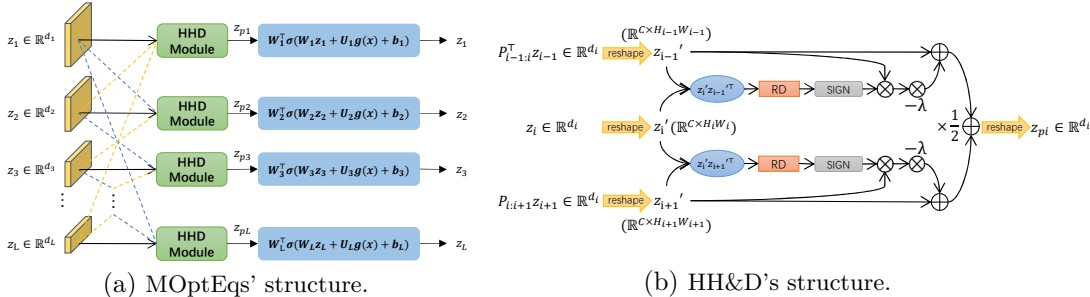

(a) MOptEqs' structure.        (b) HH&D's structure.

Figure 1: The structure of the MOptEqs and HH&D Module. Dotted lines in the figure denotes the upsampling, downsampling or identity operators to suit the size of each branch, $\otimes$ denotes multiplication operator, $\oplus$ denotes addition operator, **RD** operator processes a matrix by removing its diagonal, **SIGN** operator convert each element of a matrix to its sign and the $d_i = CH_iW_i$ with $C$ is the channel number and $H_i, W_i$ are the height and width of the feature map.

*The equilibrium points for the above formula are also the first-order stationary points for the optimization problem Eqn(5).*

The proof of the proposition is listed in Appendix A.3. In this manner, we obtain the proposed architecture MOptEqs, its forward propagation is equivalent to find the equilibrium points of Eqn(6), and such process can also be regarded as solving the stationary points of the optimization problem (5). The whole structure is also shown in Figure 1 (a) and we also draw figures to illustrate the practical process of $h_i$ (HH&D) on its right (Figure 1 (b)). We also evaluate the effectiveness of such modelling in Section. 3.2's experiments.

## 2.2 The proposed Perturbation Enhanced Strategy for MOptEqs

Researches (Kong et al., 2020; Xie et al., 2020; Tang et al., 2021) have discovered that small perturbations can enhance the generalization abilities and varieties of branches for neural architectures. Most efficient works use adversarial perturbations based on gradients (Xie & Yuille, 2020) for such a trick. However, such a strategy will slow the training process since they need at least one more back-propagation in each training step for perturbations, which is time-consuming. Unlike these methods, we can acquire more computation efficient perturbations because our models can be formulated as minimizing an objective function we designed. Then we can replace the maximum problem (details in Appendix A.2) for adversarial perturbation with maximizing our architecture's hidden objective function $G$, since we assume MOptEq's performance will go worse if $G(\mathbf{z}; g(\mathbf{x}))$ changes a lot at $g(\mathbf{x})$'s neighborhoods since the forward propagation of MOptEqs is minimization $G(\mathbf{z}; g(\mathbf{x}))$.

**Assumption 1** *For a well trained OptEqs with its objective function denoted as $G(\mathbf{z}; g(\mathbf{x}))$ and a natural sample $\mathbf{x}$ which can be correctly classified. If an input perturbation $\|\delta\|_\infty \leq \epsilon$ can cause one of the following changes:*

$$|G(\mathbf{z}'^*; g(\mathbf{x}) - \delta) - G(\mathbf{z}^*; g(\mathbf{x}))| \geq L_1\epsilon,$$
$$\|\mathbf{z}'^* - \mathbf{z}^*\|_2 \geq L_2\epsilon.$$

*with $L_1, L_2 > 1$, it may lead the OptEqs to the wrong results with high probability.*

For the second part of the assumption, it's common sense and widely used in the analysis of adversarial robustness (Zhang et al., 2021; Li et al., 2020). With the above assumption, we can generate the perturbations by maximizing the objective function $G(\mathbf{z}; g(\mathbf{x}))$ for our MOptEqs. Since our MOptEqs can be regarded as a kind of "implicit" ensembling that parallels several equilibrium models, we decide to inject the perturbations acquired for the prior branch to the posterior one, like boosting to enhance the performance.

Then the hidden objective function of MOptEqs with $G(\mathbf{z}; g(\mathbf{x}))$ is shown as follows,

$$
\min_{\mathbf{z}_1,...,\mathbf{z}_L} G(\mathbf{z}_1,...,\mathbf{z}_L; g(\mathbf{x})) = \min_{\mathbf{z}_1,...,\mathbf{z}_L} \sum_{i=1}^{L} \left[ \mathbf{1}^\top f(\mathbf{W}_i^{-1\top}\mathbf{z}_i) - \left\langle \mathbf{U}_i(g(\mathbf{x}) - \delta_{i-1}) + \mathbf{b}_i, \mathbf{W}_i^{-1\top}\mathbf{z} \right\rangle \right.
$$
$$
\left. + \frac{1}{2} \left( \lambda \mathcal{D}(\mathbf{z}_i, \mathbf{z}_{i+1}) + \|\mathbf{W}^{-1\top}\mathbf{z}_i\|^2 - \mathcal{H}(\mathbf{z}_i, \mathbf{z}_{i+1}) \right) \right],
$$
$$
s.t. \quad \delta_i = \operatorname*{argmax}_{\|\delta_i\|_\infty \le \epsilon} \mathbf{1}^\top f(\mathbf{W}_i^{-1\top}\mathbf{z}_i) - \left\langle \mathbf{U}_i(g(\mathbf{x}) - \delta_i) + \mathbf{b}_i, \mathbf{W}_i^{-1\top}\mathbf{z}_i \right\rangle \quad \text{for } i \in [1, L].
$$

where $\delta_0 = 0$. And the problem becomes a bilevel optimization problem. The solution to the lower-level problem is $\delta_i = \epsilon \mathbf{sign}(\mathbf{U}_i^\top \mathbf{W}_i^{-1\top}\mathbf{z}_i)$. Since the perturbation can be obtained by feeding the output of the activation layer $\sigma(\mathbf{W}\mathbf{z} + \mathbf{U}g(\mathbf{x}) + \mathbf{b})$ to the transposed convolution layer with weight $\mathbf{U}_i$, we call it reconstructed perturbation. Compared with the adversarial perturbations based on gradients, our perturbations can be acquired directly by matrix-vector multiplication instead of iteratively built by gradients. Thus, such a process does not require much computation cost. Following the above steps, we obtain a harmful perturbed direction for the prior branches and then added them to their posterior branches. We note that our perturbation is added to the input feature $g(\mathbf{x})$ instead of the raw input.

**Proposition 2** *For a natural sample $\mathbf{x}$ and a well trained OptEqs*

$$
\mathbf{z}^* = \mathbf{W}^\top \sigma(\mathbf{W}\mathbf{z}^* + \mathbf{U}g(\mathbf{x}) + \mathbf{b})
$$

*with its hidden objective function denoted as $G(\mathbf{z}; g(\mathbf{x}))$, for $g'(\mathbf{x}) = g(\mathbf{x}) - \delta$ and $\delta = \epsilon \mathbf{sign}(\mathbf{U}_i^\top \mathbf{W}_i^{-1\top}\mathbf{z}_i)$, which obeys the constraint $\|\delta\|_\infty \le \epsilon$. $\mathbf{z}'^*$ and $\mathbf{z}^*$ are defined as follows:*

$$
\mathbf{z}'^* = \operatorname*{argmin}_{\mathbf{z}} G(\mathbf{z}; g'(\mathbf{x})),
$$
$$
\mathbf{z}^* = \operatorname*{argmin}_{\mathbf{z}} G(\mathbf{z}; g(\mathbf{x})).
$$

*If the $\|\mathbf{W}\|_2, \|\mathbf{U}\|_2 \le 1$ and $\frac{1}{N}\|\mathbf{U}^\top \mathbf{W}^{-1\top}\mathbf{z}^*\|_1 \gg \epsilon$, where $N$ is the element number of $g(\mathbf{x})$, then*

$$
\max\left\{ G(\mathbf{z}'^*; g(\mathbf{x}) - \delta) - G(\mathbf{z}^*; g(\mathbf{x})), \sqrt{N}\epsilon\|\mathbf{z}'^* - \mathbf{z}^*\|_2 \right\} \ge \frac{1}{2}\left\{ \epsilon\|\mathbf{U}^\top \mathbf{W}^{-1\top}\mathbf{z}^*\|_1 - N\epsilon^2 \right\}.
$$

The proposition demonstrates if we choose the perturbed direction to be $\epsilon \mathbf{sign}(\mathbf{U}^\top \mathbf{W}^{-1\top}\mathbf{z})$, at least one of the changes for the $G$ or $\mathbf{z}^*$ will be around $\epsilon\|\mathbf{U}^\top \mathbf{W}^{-1\top}\mathbf{z}^*\|_1$ and implies each branch may perform bad on the perturbed data according to our assumptions. With the experiments in Section.3.2, we can conclude that our reconstructed perturbation is useful and our assumption is reasonable. Like boosting strategy, we can feed the perturbed data for the prior branch to its posterior branch to enhance the performance; we call such method as the **Perturbation Enhanced strategy** (PE). Following experiments also show that PE can indeed enhance the performance of our MOptEqs.

## 2.3 Model Optimization and Forward Convergence

**Forward Propagation and Implicit Differentiation.** Like other equilibrium models, the forward propagation procedure is solving the given equilibrium functions Eqn 7. For this count, our model also enjoys the constant memory cost advantages as other DEQs. In our work, run root-finding algorithms to solve the roots $\tilde{\mathbf{z}}^* = [\mathbf{z}_1^*, ...\mathbf{z}_L^*]$ of the following problem to reach the equilibrium states, which is the same as MDEQ (shown in Appendix A.5):

$$
\nabla_{\tilde{\mathbf{z}}} G(\mathbf{z}_1, ..., \mathbf{z}_L; g(\mathbf{x})) = 0. \tag{7}
$$

As for the backward propagation, we also adopt the implicit differentiation method widely used in (Bai et al., 2020; 2019; Chen et al., 2018). Instead of tracing the gradients during the forward propagation, the implicit differentiation method directly backpropagates through the equilibrium state using the Jacobian of $\mathcal{T}_\theta = \nabla_{\tilde{\mathbf{z}}} G + \tilde{\mathbf{z}}$ at $\tilde{\mathbf{z}}^*$. For a given loss $l = \mathcal{L}(\tilde{\mathbf{z}}^*, \mathbf{y})$ (where $\mathbf{y}$ is the target) and the gradients can be written as

$$
\frac{\partial l}{\partial(\cdot)} = \frac{\partial l}{\partial \tilde{\mathbf{z}}^*}(\mathbf{I} - \mathbf{J}_{\mathcal{T}_\theta}|_{\tilde{\mathbf{z}}^*})^{-1}\frac{\partial \mathcal{T}_\theta}{\partial(\cdot)}. \tag{8}
$$

| Model | Model Size | Accuracy |
|---|---|---|
| Neural ODEs | 172K | 53.7% |
| Aug. Neural ODEs | 172K | 60.6% |
| Single-tire MonDEQ | 854K | 82.5% |
| Deep OptEqs[1] | 199K | 87.4% |
| Parallel-OptEqs | 193K | 87.4% |
| POptEqs(sum) | 193K | 88.4% |
| POptEqs(conv) | 276K | 88.9% |
| MOptEqs (w/o PE) | 193K | 89.1% |
| MOptEqs | 193K | **89.5**% |

(a) Comparison of the models with single-scales

| Model | Model Size | Accuracy |
|---|---|---|
| ResNet-18 | 10M | $92.9 \pm 0.2\%$ |
| MonDEQ | 1M | 89.7% |
| MDEQ | 2.53M | $92.6 \pm 0.2\%$ |
| MDEQ | 10M | $93.8 \pm 0.3\%$ |
| MOptEqs | 0.48M | $\mathbf{92.9 \pm 0.2}\%$ |
| MOptEqs | 1.9M | $\mathbf{94.0 \pm 0.1}\%$ |
| MOptEqs | 8.1M | $\mathbf{94.6}\%$ |

(b) Comparison of the models with multiple-scales. MonDEQ here is the multi-tired monotone DEQ.

Table 1: Evaluation on CIFAR-10 for different models. "Parallel-OptEqs"(POptEqs) means our model is built without utilizing HH&D fusion (Stated in Appendix A.1), which is formed by paralleling several OptEqs with the method stated in the brackets for fusion. "w/o PE" means MOptEqs is trained without our PE strategy.

And the calculation of $\frac{\partial l}{\partial \tilde{\mathbf{z}}^*}(\mathbf{I} - \mathbf{J}_{\mathcal{T}_\theta}|\tilde{\mathbf{z}}^*)^{-1}$ is equivalent to solve root $\mathbf{m}$ for the following equation:

$$\mathbf{m}(\mathbf{I} - \mathbf{J}_{\mathcal{T}_\theta}|\tilde{\mathbf{z}}^*) + \frac{\partial l}{\partial \tilde{\mathbf{z}}^*} = 0. \tag{9}$$

As for the root-finding algorithm, we can use Broyden method (Broyden, 1965), Anderson Method (Anderson, 1965; Bai et al., 2021) or other root-finding methods to solve Eqn(7) for the equilibrium state and Eqn(9) for the backward gradient.

**Forward Convergence.** Like other implicit models, we make some constraints on parameters in order to make the whole MOptEqs $\mathcal{T}_\Theta(\tilde{\mathbf{z}}; \mathbf{x})$ ($\tilde{\mathbf{z}}$ here is $\{\mathbf{z}_i\}_{i=1}^L$) be a contractive mapping. For the MOptEqs without considering HH&D (Eqn 10), the models can easily converge with $\|\mathbf{W}_i\|_2 \leq \zeta < 1$. But since we use the HH&D module, we need to choose proper $\lambda$ to ensure the convergence, around or less than $\frac{1}{C}$ will be appropriate. We also conduct experiments to verify the convergence in Appendix A.7.

## 3 EXPERIMENTAL RESULTS

In this section, we conduct experiments for the image classification tasks on CIFAR-10,CIFAR-100 Krizhevsky et al. (2009) and ImageNette implemented on the Py-Torch (Paszke et al., 2017) platform to demonstrate the effectiveness of our model. Details are listed in Appendix A.6.

### 3.1 COMPARISON OF PRIOR IMPLICIT MODELS

In this part, we decide to verify the superiority of MOptEqs in two aspects. First, we build the single-scale MOptEqs and compare the empirical results with other single-scale implicit models. And then we compare the experimental results with prior implicit models that utilize the multi-scale inputs like MDEQs, the state-of-the-art implicit model with multi-resolution.

**Implicit Models with Single View.** In this part, we compare our MOptEqs with other single-view implicit models with comparable model sizes. Like OptEqs who uses three blocks for their experiment, we first construct our small MOptEqs with three branches whose outputs share the same size and the channel number $C = 32$. Results in Table.1 (a) demonstrates that our MOptEqs structure can even enhance the performance for recognition tasks under the single-view cases. Compared to the multi-branch model without fusion (POptEqs), our MOptEqs' HH&D can efficiently utilize the relationships between different branches and lead the model to better performance. Our superiority also holds compared with former multi branches fusion methods: "sum" (non-parametric module) and "conv" (parametric module). Meanwhile, we can also conclude the effectiveness of our perturbation enhanced strategy from the table since its performance is the best.

| Model | Model Size | Accuracy |
|-------|-----------|----------|
| MDEQ | 2.6M | $70.8 \pm 0.2\%$ |
|       | 11M  | $72.4 \pm 0.2\%$ |
| MOptEqs | 1.9M | $\mathbf{73.4 \pm 0.2}\%$ |
| MOptEqs | 8.1M | $\mathbf{74.7}\%$ |

(a) Evaluation on CIFAR-100 for models.

| Model | Model Size | Accuracy |
|-------|-----------|----------|
| MDEQ | 2.5M | $90.5 \pm 0.2\%$ |
|       | 10M  | $91.3 \pm 0.3\%$ |
| MOptEqs | 1.9M | $\mathbf{92.1 \pm 0.2}\%$ |
| MOptEqs | 10M  | $\mathbf{92.4}\%$ |

(b) Evaluation on Imagenette for models.

Table 2: Evaluation on CIFAR-100 and ImageNette for MDEQ and MOptEqs.

**Implicit Models with Multiple Scales.** Moreover, we conduct experiments compared with other models which handling multi-scale inputs. Like MDEQ, we construct our MOptEqs with four branches with resolution size equals to 32,16,8,4. Other details can be found in Appendix A.6. From Table.1 (b) and as shown in Table.2 (a), one can see that our MOptEqs not only outperforms the widely used explicit model ResNet-18 (He et al., 2016), but also shows better performance compared with MDEQ with fewer parameters on CIFAR, which is one of the best models. The empirical results for the multi-scale models further verify the superiority of MOptEqs and its strategy.

Apart from experiments on small images, we also conduct experiments on ImageNette, which is a subset of 10 classes from ImageNet. Compared with MDEQ, our MOptEqs consistently perform better shown in Table.2 (b). Nevertheless, as one can see from the results, the difference between MDEQ and our model becomes much bigger in the CIFAR-100 and Imagenette with the same training hyper-parameters. Such a phenomenon demonstrates that the performance of our model is much more stable than MDEQ. Apart from these experiments, we also conducted ablation studies for our models in the following section.

## 3.2 The comprehensive understanding of MOptEqs

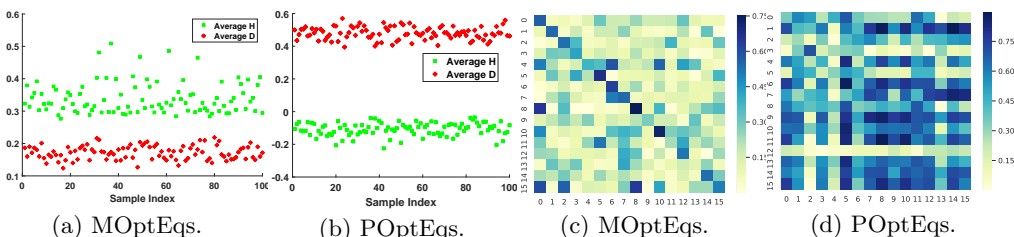

(a) MOptEqs.     (b) POptEqs.     (c) MOptEqs.     (d) POptEqs.

Figure 2: Visualization of the channels correlations for the MOptEqs and POptEqs.

**Visualization of the impact for the HH&D module.** In this section, we try to estimate the effect of our HH&D module. We finish experiments in this part using three-branch MOptEqs with 16 channels for each branch to make the visualization more clear.

We tries to verify our HH&D module's effect in two aspects. First, we plot scatter figures for the values of $\frac{\sum_{i=1}^{L} \sum_{j \in \mathcal{N}(i)} \mathcal{D}(\mathbf{z}_i, \mathbf{z}_j)}{L(C^2 - C)}$ (denoted as *Average D*) and $\frac{\sum_{i=1}^{L} \sum_{j \in \mathcal{N}(i)} \mathcal{H}(\mathbf{z}_i, \mathbf{z}_j)}{LC}$ (denoted as *Average H*) for 100 randomly chosen samples ($\mathcal{N}(i)$ denotes neighboring branch of the $i$-th branch). Since the hidden optimization problem is to maximizing $\mathcal{H}$ while minimizing $\mathcal{D}$, we find that our MOptEqs (Figure 2(a)) can induce the output features to reach such goal compared with POptEqs (MOptEqs without HH&D) (Figure 2(b)).

Furthermore, we also plot heatmaps of the first two branches' correlation score ($|\mathbf{vec}^{-1}(\mathbf{z}_2)\mathbf{vec}^{-1}(\mathbf{z}_1)^\top| \in \mathbb{R}^{16 \times 16}$) for a randomly chosen sample shown in Figure 2(c)(d). One can see that MOptEqs' heatmap (Figure 2(c)) is more likely to be a diagonal matrix while heatmap Figure 2(d) for POptEqs looks random. The heatmap shows that our HH&D can induce the model to perform as our demand, which means corresponding channels for adjacent branches can be more correlated, while the dis-corresponding channels are unrelated due to our architecture design. The visualizations and former results on different datasets verify the effectiveness of our modelling and our HH&D modules' design.

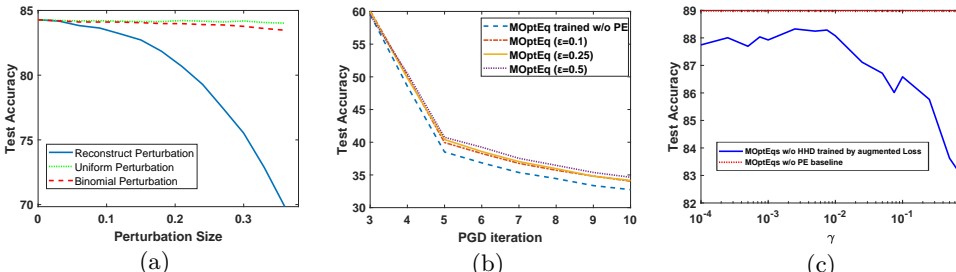

Figure 3: (a) Test accuracy changes with respect to the perturbation size for different perturbed directions. (b) Test Accuracy for small MOptEqs (w. and w/o. PE) under PGD attack with different inner iterations. (c) Plot of the accuracies for the small MOptEqs without HH&D trained by the extended loss with respect to various $\gamma$.

**Evaluation of the reconstructed perturbation.** We compare our perturbation with randomly generated ones in a one-branch well-trained MOptEqs to validate its effectiveness. We first feed a natural sample $\mathbf{x}$ to the model and acquire output $\mathbf{z}$ for $\mathbf{x}$ and generate our reconstructed perturbation using such output and then feed it to the model. Furthermore, we add the perturbations generated by Binomial distribution ($\mathbb{P}(\delta_i = \epsilon) = \mathbb{P}(\delta_i = -\epsilon) = 0.5$) and Uniform distribution $\mathcal{U}[-\epsilon, \epsilon]$ to input features $g(\mathbf{x})$ for comparison. The result of each distribution is averaged for five trials. Figure 3(a) shows our perturbation generated by maximizing $G$ is much more effective, which validates the effectiveness of our reconstructed perturbations and the rationality of our assumptions and analysis in Section.2.2.

**Robustness of our MOptEqs trained by PE strategy.** In addition to improving the generalization abilities for models as shown in Sec.3.1, perturbations with proper size may also enhance the robustness of our MOptEqs, shown in Figure 3(b). We conduct experiments on small MOptEqs (the same setting as Sec.3.1's single-view model). Our perturbation enhanced strategy goes stronger as $\epsilon$ increases. The figure shows that accuracy for the model trained naturally drops quicker than trained by PE strategy. Overall, trained with appropriate reconstructed perturbation can partly improve the robustness of our MOptEqs.

**MOptEqs vs. adding Regularizers in training loss.** Apart from our HH&D modules, adding $\mathcal{D}$ and $\mathcal{H}$ to the training loss is one of the most popular methods for obtaining outputs with certain properties. In order to demonstrate the superiority of our module empirically, we conduct experiments for our MOptEqs trained by cross-entropy and POptEqs (MOptEqs (w/o. HH&D)) trained by the cross-entropy loss adding $\gamma \sum_{i=1}^{L} (\mathcal{H}(\mathbf{z}_i, \mathbf{z}_{i+1}) - \lambda \mathcal{D}(\mathbf{z}_i, \mathbf{z}_{i+1}))$ as regularizers (we call it augmented loss). Then we drew its test accuracy for different $\gamma$ trained by the augmented loss ($\lambda$ is the same as ours) shown in Figure 3(c).

With proper $\gamma$, POptEqs trained by additional loss can perform better than $\gamma = 0$. Such phenomenon demonstrates the effectiveness of our HH&D Modelling. Furthermore, the figure demonstrates the superiority of our model since the traditional method is consistently worse than ours. We left some other explorations for our model in the Appendix.

## 4 CONCLUSION

We introduce the multi-branch optimization induced equilibrium models (MOptEqs), a new extension of OptEqs that can utilize multiscale information for recognition tasks and retain its ability to recover to an optimization problem whose solution is equivalent to the equilibrium states of our model. The model architecture is designed based on modelling the hidden objective function for the multi-resolution recognition task. Furthermore, we also propose a new strategy inspired by our understandings of the hidden optimization problem. The empirical results show the advantages of our proposed methods. The success of our HH&D module and PE strategy demonstrates the deep link between the optimization problem and neural architecture and may motivate further explorations.

ACKNOWLEDGMENTS

Yisen Wang is partially supported by the National Natural Science Foundation of China under Grant 62006153, Project 2020BD006 supported by PKU-Baidu Fund, and Open Research Projects of Zhejiang Lab (No. 2022RC0AB05).

Zhouchen Lin is supported by the NSF China (No. 61731018), NSFC Tianyuan Fund for Mathematics (No. 12026606), Project 2020BD006 supported by PKU-Baidu Fund, and Qualcomm.

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

# A APPENDIX

## A.1 PARALLELING MULTI-BRANCH OPTEQS

Experiments show that even a simple one-layer OptEqs module enjoys impressive generalization abilities. If we view a simple one-layer OptEqs module as a powerful feature generator, we can use an "ensemble" scheme on the DEQs by paralleling several DEQ modules. Then we get the multi-branch implicit structure (formulated as Eqn 10), which is capable of utilizing samples of multi-resolution but also retains the capability of recovering to an optimization problem.

$$
\begin{aligned}
\mathbf{z}_1^* &= \mathbf{W}_1^\top \sigma(\mathbf{W}_1 \mathbf{z}_1^* + \mathbf{U}_1 g(\mathbf{x}) + \mathbf{b}_1), \\
&\quad \ldots\ldots \\
\mathbf{z}_L^* &= \mathbf{W}_L^\top \sigma(\mathbf{W}_L \mathbf{z}_L^* + \mathbf{U}_L g(\mathbf{x}) + \mathbf{b}_L),
\end{aligned}
\tag{10}
$$

where $\mathbf{z}_i^*$ denotes the equilibrium outputs of $i$-th branch, we call this structure Parallel-OptEqs (POptEqs). Like (Xie et al., 2021) stated for OptEqs, the POptEqs can also be depicted as solving the following problem (Eqn 11) if $\mathbf{W}_i$ is invertible and $\sigma(\cdot)$ is activation function,

$$
\min_{\mathbf{z}_1,\ldots,\mathbf{z}_L} G(\mathbf{z}_1,\ldots,\mathbf{z}_L; g(\mathbf{x})) = \min_{\mathbf{z}_1,\ldots,\mathbf{z}_L} \sum_{i=1}^L \left[ \mathbf{1}^\top f(\mathbf{W}_i^{-1\top} \mathbf{z}_i) - \left\langle \mathbf{U}_i g(\mathbf{x}) + \mathbf{b}_i, \mathbf{W}_i^{-1\top} \mathbf{z} \right\rangle \right.
$$
$$
\left. + \frac{1}{2} \|\mathbf{W}_i^{-1\top} \mathbf{z}_i\|^2 - \frac{1}{2} \|\mathbf{z}_i\|^2 \right],
\tag{11}
$$

where $g(\mathbf{x})$ is the input feature with raw input $\mathbf{x}$, $\mathbf{z}_i \in \mathbb{R}^{d_i \times 1}$ and $f$ added for the activation function. For example $f$ is the positive orthant $f(x) = \mathbb{I}\{x \geq 0\}$ if we use ReLU activation, since $(I + \partial f)^{-1}$ is the ReLU activation function, other activation functions can be found in Li et al. (2019).The calculation of the above structure (Eqn 10) is equivalent to solving the stationary points $\nabla_{\mathbf{z}_i} G = 0$.

## A.2 ADVERSARIAL PERTURBATIONS.

Adversarial perturbations aim to cheat well-trained neural networks with small or unnoticeable changes on natural images. The adversarial perturbations can be obtained by solving the following maximum problem for input $\mathbf{x}_{nat}$ and label $y$:

$$
\max_{\delta \in \mathcal{C}} \mathcal{L}(f_{NN}(\mathbf{x}_{nat} + \delta); y),
$$

where $\mathcal{C}$ is the feasible set of $\delta$ in case the change is too large and usually chosen to be an l-infinite ball. Due to the neural network is a non-convex function for $\delta$, obtaining the perturbation is not easy. The most popular method is the PGD-$k$ method which solves the above problem by implementing the projected gradient method by $k$ times. Although the performance of such a method is satisfactory in most cases, it makes the whole training procedure $k + 1$ times slower than the origin. In our work, we replace the above objective function with our model's hidden objective function. Since the hidden optimization problem is convex with respect to the perturbation, we can get the perturbation much easier. Moreover, such perturbation can also enhance the generalization abilities, as shown in the experiments.

## A.3 PROOFS FOR PROPOSITION.1

In this section, we are going to prove that the equilibrium equation Eqn 1 is also the first order stationary points for the optimization problem 5.

**Proof 1** *The equilibrium equations Eqn 1 in the proposition can be separated into different branches as follows:*

$$\mathbf{z}_1 = \mathbf{W}_1^\top \sigma\left(\mathbf{W}_1 h_1\left(\mathbf{z}_1, \mathbf{z}_L, \mathbf{z}_2\right) + \mathbf{U}_1 g(\mathbf{x}) + \mathbf{b}_1\right),$$

$$\cdots\cdots$$

$$\mathbf{z}_i = \mathbf{W}_i^\top \sigma\left(\mathbf{W}_i h_i\left(\mathbf{z}_i, \mathbf{z}_{i-1}, \mathbf{z}_{i+1}\right) + \mathbf{U}_i g(\mathbf{x}) + \mathbf{b}_i\right), \tag{12}$$

$$\cdots\cdots$$

$$\mathbf{z}_L = \mathbf{W}_L^\top \sigma\left(\mathbf{W}_L h_L\left(\mathbf{z}_L, \mathbf{z}_{L-1}, \mathbf{z}_1\right) + \mathbf{U}_L g(\mathbf{x}) + \mathbf{b}_L\right),$$

*where $h_i : \mathbb{R}^{d_i} \times \mathbb{R}^{d_{i+1}} \times \mathbb{R}^{d_{i-1}} \to \mathbb{R}^{d_i}$ can be defined as:*

$$
\begin{aligned}
h_i(\mathbf{z}_i, \mathbf{z}_{i-1}, \mathbf{z}_{i+1}) = &\frac{1}{2}(\mathbf{P}_{i:i+1}\mathbf{z}_{i+1} + \mathbf{P}_{i-1:i}^\top \mathbf{z}_{i-1}) \\
&- \frac{\lambda}{2}\mathbf{vec}\left(\mathbf{vec}^{-1}(\mathbf{P}_{i:i+1}\mathbf{z}_{i+1})\mathbf{sign}\left[\mathbf{M}_{i:i+1} - \mathbf{diag}(\mathbf{M}_{i:i+1})\right]\right) \\
&- \frac{\lambda}{2}\mathbf{vec}(\mathbf{vec}^{-1}\left(\mathbf{P}_{i-1:i}^\top \mathbf{z}_{i-1}\right)\mathbf{sign}\left[\mathbf{M}_{i-1:i}^\top - \mathbf{diag}(\mathbf{M}_{i-1:i})\right]),
\end{aligned} \tag{13}
$$

*where*

$$\mathbf{M}_{i:i+1} = \mathbf{vec}^{-1}(\mathbf{P}_{i:i+1}\mathbf{z}_{i+1})^\top \mathbf{vec}^{-1}(\mathbf{z}_i).$$

*First, we need to prove that*

$$h(\mathbf{z}_i, \mathbf{z}_{i-1}, \mathbf{z}_{i+1}) = -\frac{1}{2}\nabla_{\mathbf{z}_i}\sum_{i=1}^{L}[\lambda\mathcal{D}(\mathbf{z}_i, \mathbf{z}_{i+1}) - \mathcal{H}(\mathbf{z}_i, \mathbf{z}_{i+1})] \tag{14}$$

*The derivatives for the right term can be divided as the derivatives for $\mathcal{H}$ and $\mathcal{D}$:*

$$
\begin{aligned}
\nabla_{\mathbf{z}_i}\sum_{k=1}^{L}\mathcal{H}(\mathbf{z}_k, \mathbf{z}_{k+1}) &= \nabla_{\mathbf{z}_i}\sum_{k=1}^{L}\mathbf{z}_k^\top \mathbf{P}_{k:k+1}\mathbf{z}_{k+1} \\
&= \nabla_{\mathbf{z}_i}(\mathbf{z}_i^\top \mathbf{P}_{i:i+1}\mathbf{z}_{i+1} + \mathbf{z}_{i-1}^\top \mathbf{P}_{i-1:i}\mathbf{z}_i) \\
&= \mathbf{P}_{i:i+1}\mathbf{z}_{i+1} + \mathbf{P}_{i-1:i}^\top \mathbf{z}_{i-1}
\end{aligned}
$$

*which is the first term of $h_i$. Then we are going to calculate the derivatives for $\mathcal{D}$:*

$$
\begin{aligned}
\nabla_{\mathbf{z}_i}\sum_{k=1}^{L}\mathcal{D}(\mathbf{z}_k, \mathbf{z}_{k+1}) &= \nabla_{\mathbf{z}_i}\sum_{k=1}^{L}\sum_{k_1=1}^{c}\sum_{k_2\neq k_1}^{c}|\mathbf{vec}^{-1}(\mathbf{z}_k)^{(k_1)\top}\mathbf{vec}^{-1}(\mathbf{P}_{k:k+1}\mathbf{z}_{k+1})^{(k_2)}| \\
&= \nabla_{\mathbf{z}_i}\left(\sum_{k_1=1}^{c}\sum_{k_2\neq k_1}^{c}|\mathbf{vec}^{-1}(\mathbf{z}_i)^{(k_1)\top}\mathbf{vec}^{-1}(\mathbf{P}_{i:i+1}\mathbf{z}_{i+1})^{(k_2)}| \right. \\
&\quad\left. + \sum_{k_1=1}^{c}\sum_{k_2\neq k_1}^{c}|\mathbf{vec}^{-1}(\mathbf{z}_{i-1})^{(k_1)\top}\mathbf{vec}^{-1}(\mathbf{P}_{i-1:i}\mathbf{z}_i)^{(k_2)}|\right) \\
&= \mathbf{vec}\left[\nabla_{\mathbf{vec}^{-1}(\mathbf{z}_i)}\left(\sum_{k_1=1}^{c}\sum_{k_2\neq k_1}^{c}|\mathbf{vec}^{-1}(\mathbf{z}_i)^{(k_1)\top}\mathbf{vec}^{-1}(\mathbf{P}_{i:i+1}\mathbf{z}_{i+1})^{(k_2)}| \right.\right. \\
&\quad\left.\left. + \sum_{k_1=1}^{c}\sum_{k_2\neq k_1}^{c}|\mathbf{vec}^{-1}(\mathbf{z}_{i-1})^{(k_1)\top}\mathbf{vec}^{-1}(\mathbf{P}_{i-1:i}\mathbf{z}_i)^{(k_2)}|\right)\right]
\end{aligned}
$$

*with the following equation holds for the sampling matrix $\mathbf{P}_{i-1:i}$ and sampling is done for each channel independently:*

$$\mathbf{vec}^{-1}(\mathbf{z}_{i-1})^{(k_1)\top}\mathbf{vec}^{-1}(\mathbf{P}_{i-1:i}\mathbf{z}_i)^{(k_2)} = \mathbf{vec}^{-1}(\mathbf{P}_{i-1:i}^\top \mathbf{z}_{i-1})^{(k_1)\top}\mathbf{vec}^{-1}(\mathbf{z}_i)^{(k_2)}$$

*The equation can be formulated as:*

$$\nabla_{\mathbf{z}_i} \frac{1}{2} \sum_{k=1}^{L} \mathcal{D}(\mathbf{z}_k, \mathbf{z}_{k+1}) = \mathbf{vec} \left[ \nabla_{\mathbf{vec}^{-1}(\mathbf{z}_i)} \left( \sum_{k_1=1}^{c} \sum_{k_2 \neq k_1}^{c} |\mathbf{vec}^{-1}(\mathbf{z}_i)^{(k_1)\top} \mathbf{vec}^{-1}(\mathbf{P}_{i:i+1}\mathbf{z}_{i+1})^{(k_2)}| \right. \right.$$

$$\left. \left. + \sum_{k_1=1}^{c} \sum_{k_2 \neq k_1}^{c} |\mathbf{vec}^{-1}(\mathbf{P}_{i-1:i}^{\top}\mathbf{z}_{i-1})^{(k_1)\top} \mathbf{vec}^{-1}(\mathbf{z}_i)^{(k_2)}| \right) \right]$$

*Since*

$$\nabla_{\mathbf{vec}^{-1}(\mathbf{z}_i)} \sum_{k_1=1}^{c} \sum_{k_2 \neq k_1}^{c} |\mathbf{vec}^{-1}(\mathbf{z}_i)^{(k_1)\top} \mathbf{vec}^{-1}(\mathbf{P}_{i:i+1}\mathbf{z}_{i+1})^{(k_2)}|$$

$$= \left[ \nabla_{\mathbf{vec}^{-1}(\mathbf{z}_i)^{(1)}} \left( \sum_{k_1=1}^{c} \sum_{k_2 \neq k_1}^{c} |\mathbf{vec}^{-1}(\mathbf{z}_i)^{(k_1)\top} \mathbf{vec}^{-1}(\mathbf{P}_{i:i+1}\mathbf{z}_{i+1})^{(k_2)}| \right), ..., \nabla_{\mathbf{vec}^{-1}(\mathbf{z}_i)^{(C)}}(...) \right]$$

$$= \left[ \sum_{k_2 \neq 1}^{L} \mathrm{sign}(\mathbf{vec}^{-1}(\mathbf{z}_i)^{(1)\top} \mathbf{vec}^{-1}(\mathbf{P}_{i:i+1}\mathbf{z}_{i+1})^{(k_2)}) \mathbf{vec}^{-1}(\mathbf{P}_{i:i+1}\mathbf{z}_{i+1})^{(k_2)}, \right.$$

$$...,$$

$$\left. \sum_{k_2 \neq C}^{L} \mathrm{sign}(\mathbf{vec}^{-1}(\mathbf{z}_i)^{(C)\top} \mathbf{vec}^{-1}(\mathbf{P}_{i:i+1}\mathbf{z}_{i+1})^{(k_2)}) \mathbf{vec}^{-1}(\mathbf{P}_{i:i+1}\mathbf{z}_{i+1})^{(k_2)} \right]$$

$$= \mathbf{vec}^{-1}(\mathbf{P}_{i:i+1}\mathbf{z}_{i+1})\mathbf{sign}[\mathbf{vec}^{-1}(\mathbf{P}_{i:i+1}\mathbf{z}_{i+1})^{\top} \mathbf{vec}^{-1}(\mathbf{z}_i) - \mathbf{diag}(\mathbf{vec}^{-1}(\mathbf{P}_{i:i+1}\mathbf{z}_{i+1})^{\top} \mathbf{vec}^{-1}(\mathbf{z}_i))]$$

$$= \mathbf{vec}^{-1}(\mathbf{P}_{i:i+1}\mathbf{z}_{i+1})\mathbf{sign}[\mathbf{M}_{i:i+1} - \mathbf{diag}(\mathbf{M}_{i:i+1})],$$

*In the same manner, we can get,*

$$\nabla_{\mathbf{vec}^{-1}(\mathbf{z}_i)} \sum_{k_1=1}^{c} \sum_{k_2 \neq k_1}^{c} |\mathbf{vec}^{-1}(\mathbf{P}_{i-1:i}^{\top}\mathbf{z}_{i-1})^{(k_1)\top} \mathbf{vec}^{-1}(\mathbf{z}_i)^{(k_2)}|$$

$$= \mathbf{vec}^{-1}(\mathbf{P}_{i-1:i}^{\top}\mathbf{z}_{i-1})\mathbf{sign}[\mathbf{M}_{i-1:i}^{\top} - \mathbf{diag}(\mathbf{M}_{i-1:i})]$$

*Taking all the terms together, we've proved the correctness of Eqn 14. Then we can proved the relationships between Eqn 12 and the first-order stationary points for Eqn 5. Take $\mathbf{z}_i$ for an example:*

$$0 = \nabla_{\mathbf{z}_i} \sum_{i=1}^{L} \left[ \mathbf{1}^{\top} f(\mathbf{W}_i^{-1\top}\mathbf{z}_i) - \langle \mathbf{U}_i g(\mathbf{x}) + \mathbf{b}_i, \mathbf{W}_i^{-1\top}\mathbf{z} \rangle + \frac{1}{2} \left( \lambda \mathcal{D}(\mathbf{z}_i, \mathbf{z}_{i+1}) + \|\mathbf{W}^{-1\top}\mathbf{z}_i\|^2 - \mathcal{H}(\mathbf{z}_i, \mathbf{z}_{i+1}) \right) \right]$$

$$0 = \mathbf{W}_i^{-1}(\mathbf{I} + \partial f)(\mathbf{W}_i^{-1\top}\mathbf{z}_i) - \mathbf{W}_i^{-1}(\mathbf{U}_i g(\mathbf{x}) + \mathbf{b}_i) - h(\mathbf{z}_i, \mathbf{z}_{i-1}, \mathbf{z}_{i+1})$$

*Then we can get the equilibrium equation:*

$$\mathbf{z}_i = \mathbf{W}_i^{\top} \sigma(\mathbf{W}_i h_i(\mathbf{z}_i, \mathbf{z}_{i-1}, \mathbf{z}_{i+1}) + \mathbf{U}_i g(\mathbf{x}) + \mathbf{b}_i),$$

*the proof is the same for all $i$. In this manner, we proved the relations between the structure and the optimization problems. The proof for the proposition is complete.*

## A.4 Proofs for Proposition.2

**Proof 2** *From the preliminaries above, the difference between $G(\mathbf{z}'^*; g(\mathbf{x}) - \delta)$ and $G(\mathbf{z}^*; g(\mathbf{x}))$ with $\delta = \epsilon SIGN(\mathbf{U}^{\top}\mathbf{W}^{-1\top}\mathbf{z}^*)$ is as follows:*

$$G(\mathbf{z}'^*; g(\mathbf{x}) - \delta) - G(\mathbf{z}^*; g(\mathbf{x})) = G(\mathbf{z}'^*; g(\mathbf{x})) - G(\mathbf{z}^*; g(\mathbf{x})) + \langle \epsilon \mathbf{sign}(\mathbf{U}^{\top}\mathbf{W}^{-1\top}\mathbf{z}^*), \mathbf{U}^{\top}\mathbf{W}^{-1\top}\mathbf{z}'^* \rangle$$

*Since $\mathbf{z}^*$ is the minimizer of $G$ at $g(\mathbf{x})$, the above function can be converted as follows:*

$$G(\mathbf{z}'^*; g(\mathbf{x}) - \delta) - G(\mathbf{z}^*; g(\mathbf{x})) \geq \langle \epsilon \mathbf{sign}(\mathbf{U}^{\top}\mathbf{W}^{-1\top}\mathbf{z}^*), \mathbf{U}^{\top}\mathbf{W}^{-1\top}\mathbf{z}'^* \rangle$$

$$= \epsilon \|\mathbf{U}^{\top}\mathbf{W}^{-1\top}\mathbf{z}^*\|_1 - \langle \epsilon \mathbf{sign}(\mathbf{U}^{\top}\mathbf{W}^{-1\top}\mathbf{z}^*), \mathbf{U}^{\top}\mathbf{W}^{-1\top}(\mathbf{z}^* - \mathbf{z}'^*) \rangle$$

*From the structure of OptEqs, we can get:*

$$\epsilon SIGN(\mathbf{U}^\top\mathbf{W}^{-1\top}\mathbf{z}^*)^\top\mathbf{U}^\top\mathbf{W}^{-1\top}(\mathbf{z}^* - \mathbf{z}'^*)$$

$$\leq \epsilon\|\mathbf{U}^\top\mathbf{W}^{-1\top}(\mathbf{z}^* - \mathbf{z}'^*)\|_1$$

$$\leq \sqrt{N}\epsilon\|\mathbf{U}^\top\mathbf{W}^{-1\top}(\mathbf{z}^* - \mathbf{z}'^*)\|_2$$

$$\leq \sqrt{N}\epsilon\|\mathbf{U}^\top\left(\sigma(\mathbf{W}\mathbf{z}^* + \mathbf{U}g(\mathbf{x}) + \mathbf{b}) - \sigma(\mathbf{W}\mathbf{z}'^* + \mathbf{U}(g(\mathbf{x}) - \delta) + \mathbf{b})\right)\|_2$$

$$\leq \sqrt{N}\epsilon\|\sigma(\mathbf{W}\mathbf{z}^* + \mathbf{U}g(\mathbf{x}) + \mathbf{b}) - \sigma(\mathbf{W}\mathbf{z}'^* + \mathbf{U}(g(\mathbf{x}) - \delta) + \mathbf{b})\|_2$$

$$\leq \sqrt{N}\epsilon\|\mathbf{W}\mathbf{z}^* + \mathbf{U}g(\mathbf{x}) + \mathbf{b} - \mathbf{W}\mathbf{z}'^* - \mathbf{U}(g(\mathbf{x}) - \delta) - \mathbf{b}\|_2$$

$$\leq \sqrt{N}\epsilon\|\mathbf{W}(\mathbf{z}^* - \mathbf{z}'^*) + \mathbf{U}\delta\|_2$$

$$\leq \sqrt{N}\epsilon\|\mathbf{W}(\mathbf{z}^* - \mathbf{z}'^*)\|_2 + \sqrt{N}\epsilon\|\mathbf{U}\delta\|_2$$

$$\leq \sqrt{N}\epsilon\|\mathbf{z}'^* - \mathbf{z}^*\|_2 + N\epsilon^2$$

*The inequality is acquired because of the Lipshitzness of common activation function (ReLU and Leacky ReLU) and the bounds on weight matrices. With the above results, we can get:*

$$G(\mathbf{z}'^*; g(\mathbf{x}) - \delta) - G(\mathbf{z}^*; g(\mathbf{x})) \geq \epsilon\|\mathbf{U}^\top\mathbf{W}^{-1\top}\mathbf{z}^*\|_1 - \sqrt{N}\epsilon\|\mathbf{z}'^* - \mathbf{z}^*\|_2 - N\epsilon^2$$

*Then we can obtain the proposition:*

$$\max\left\{G(\mathbf{z}'^*; g(\mathbf{x}) - \delta) - G(\mathbf{z}^*; g(\mathbf{x})), \sqrt{N}\epsilon\|\mathbf{z}'^* - \mathbf{z}^*\|_2\right\} \geq \frac{1}{2}\left\{\epsilon\|\mathbf{U}^\top\mathbf{W}^{-1\top}\mathbf{z}^*\|_1 - N\epsilon^2\right\}$$

The proofs are similar when considering the HH&D modules.

### A.5 Forward Process of our MOptEqs' implicit part.

The forward propagation is illustrated as follows:

---

**Algorithm 1:** Forward Process of our MOptEqs' implicit part.

---

**Require:** Input $\mathbf{x}$, initial points $\{\mathbf{z}_i\}_{i=1}^L$, perturbation size $\epsilon$, $\delta_0 = 0$, maximum perturbation size $\epsilon$.
**Ensure:** Equilibrium points $\{\mathbf{z}_i\}_{i=1}^L$
    Calculate the input feature $g(\mathbf{x})$.
    The calculation of $\nabla_{\tilde{\mathbf{z}}}G(\tilde{\mathbf{z}}; g(\mathbf{x})) = 0$ is the same as finding the equilibrium points

$$\mathcal{T}_\theta = \nabla_{\tilde{\mathbf{z}}}G + \tilde{\mathbf{z}} = \tilde{\mathbf{z}}$$

.
    Use Broyden (Broyden, 1965) or Anderson Method (Anderson, 1965) or others to find roots of the following equation like MDEQ:

$$\mathcal{T}_\theta(\tilde{\mathbf{z}}; g(\mathbf{x})) = \tilde{\mathbf{z}}$$

    **return** $\tilde{\mathbf{z}} = \{\mathbf{z}_i^*\}_{i=1}^L$

---

And the calculation of $\mathcal{T}_\theta$ is shown as follows:

---

**Algorithm 2:** The Calculation of $\mathcal{T}_\theta = \nabla_{\tilde{\mathbf{z}}} G + \tilde{\mathbf{z}}$ at $(\tilde{\mathbf{z}}, g(\mathbf{x}))$.

---

**Require:** Input feature $g(\mathbf{x})$, initial points $\{\mathbf{z}_i\}_{i=1}^L$, perturbation size $\epsilon$, $\delta_0 = 0$, maximum perturbation size $\epsilon$.
**Ensure:** Output $\mathcal{T}_\theta = \nabla_{\tilde{\mathbf{z}}} G(\tilde{\mathbf{z}}; g(\mathbf{x})) + \tilde{\mathbf{z}}$
    **for** $i \in RANGE(1, L)$ **do**
        $k_1, k_2 \leftarrow$ neighbor of $i$ in $L$-loop
        Calculate $\mathbf{z}_{pi}$ by $h_i$ (Shown in Figure 1 (b)):
            $\mathbf{z}_{pi} \leftarrow h_i(\mathbf{z}_i, \mathbf{z}_{i-1}, \mathbf{z}_{i+1})$
        **if** Use PE Method **then**
            $\mathbf{y}_i \leftarrow \sigma(\mathbf{W}_i \mathbf{z}_{pi} + \mathbf{U}_i(g(\mathbf{x}) - \delta_{i-1}) + \mathbf{b}_i)$
            $\delta_i \leftarrow \epsilon \text{SIGN}(\mathbf{U}_i^\top \mathbf{y}_i)$
        **else**
            $\mathbf{y}_i \leftarrow \sigma(\mathbf{W}_i \mathbf{z}_{pi} + \mathbf{U}_i g(\mathbf{x}) + \mathbf{b}_i)$
        **end if**
        $\mathbf{z}_i \leftarrow \mathbf{W}_i^\top \mathbf{y}_i$
    **end for**
    **return** $\{\mathbf{z}_i^*\}_{i=1}^L$

---

## A.6 EXPERIMENTS DETAILS OF MOPTEQS FOR THE EXPERIMENTS CLASSIFICATION.

### A.6.1 CIFAR-10

For classification tasks of single-view models, we set the channel number of each branch to 32, $\lambda = 0.01$ and perturbation size $\epsilon = 0.1$ for the small MOptEqs with three branches for the single-view comparison as OptEqs. The batch size is set to be 128 for all the experiments.

As for the multi-scale models' comparison, we use four branches with each branch take inputs with resolutions equals 32, 16, 8, 4 like MDEQ, the output channel number for each branch is 256 for MOptEqs with 1.9M learnable parameters. At the same time, we set the channel number to be 128 for MOptEqs with 0.48M learnable parameters. And we set $\lambda = 0.001$ and perturbation size $\epsilon = 0.1$ for the experiments.

We use the widely used SGD algorithm for training procedure. We set weight decay to be 0.001 and the initial learning rate start from 0.1 and decay by 0.1 at $100, 150, 175$-th epoch with 200 epochs in total like others. We use the standard data augmentation for all the experiments.

Our implementation for small MDEQ only changes the channel number of MDEQ's 10M model to obtain the comparable size with our model, without changing the training settings and its hyper-parameters.

### A.6.2 CIFAR-100

Except for CIFAR-10, we also finish the experiments on CIFAR-100 classification[2] to further verify our model's effectiveness. The model we used is the same as the models used in the multi-scale comparison, only changing the output from 10 to 100 for classification. MDEQ with 11M parameter is the same model as (Bai et al., 2020) proposed for CIFAR-10 experiment with only changing the output from 10 to 100 for CIFAR-100. Moreover, the hyper-parameter setting is also the same as (Bai et al., 2020) proposed in the paper.

We use the widely used SGD algorithm for training procedure. We set weight decay to be 0.001 and the initial learning rate start from 0.1 and decay by 0.1 at $100, 150, 175$-th epoch with 200 epochs in total like others. We use the standard data augmentation for all the experiments.

---

[2]https://www.cs.toronto.edu/ kriz/cifar.html

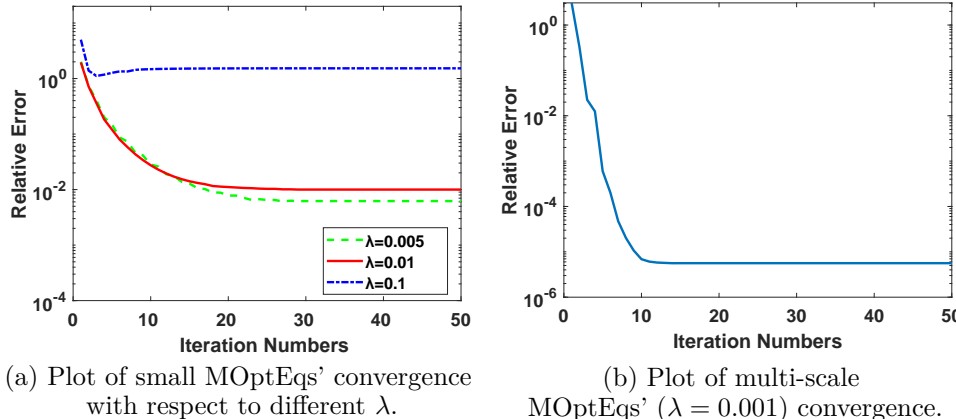

(a) Plot of small MOptEqs' convergence with respect to different $\lambda$.

(b) Plot of multi-scale MOptEqs' ($\lambda = 0.001$) convergence.

Figure 4: The convergent iterations' of our forward root-finding procedure used for our MOptEqs ($\lambda = 0.001$) forward propagation. Relative error is defined as $\frac{\|\mathcal{T}^{j+1} - \mathcal{T}^j\|_2}{\|\mathcal{T}^j\|_2}$.

### A.6.3 IMAGENETTE

Besides the classification for images with small size, we also conduct the experiments on the Imagenette[3], which is a subset of 10 classes from ImageNet[4] with 9469 training photos and its test set consists of 3925 images. Furthermore, we use the full-size version of Imagenette[5] further to verify the superiority of our models on large-scale images. The model architecture for our MOptEqs and MDEQ (for comparison) is the same as the ones used in the multi-scale models' comparison for CIFAR-10 except adding two downsampling layers in the head of models to downsample the input size from 256 to 64. As for the hyper-parameters, we only change the weight decay to be $5e - 5$, batch size to be 32 and extend the whole training epochs to 200 with 100-th, 150-th, 175-th for the learning rate decay for all the models. All the hyper-parameters for both MOptEqs and MDEQs are hardly changed during all the datasets.

### A.7 CONVERGENCE VALIDATION FOR OUR MOPTEQS.

As shown in Figure 4(a), the Anderson method can quickly find the equilibrium points for our small MOptEq used for single-view comparison. From the figure, one can see that MOptEqs with proper $\lambda$ (around or smaller than $\frac{1}{C}$, around 0.03 in this circumstance) can ensure the whole mappings be contractive. But too large $\lambda$ will make the model hard to converge or even fail.

Apart from plotting the convergent iterations' of our forward root-finding procedure for the small MOptEqs (shown in Figure 4 (a)), we also draw figures to show the convergence of the forward propagation for our multi-scale MOptEqs ($\lambda = 0.001$) used in the imagenette and CIFAR experiments as Figure 4 (b) shown. We use the pre-trained model on Imagenette whose prediction accuracy is reported in the above sections. One can see that our multi-scale MOptEqs can also quickly converge to the equilibrium points as we expected.

### A.8 ABLATION STUDIES ON THE IMPACT OF $\mathcal{D}$ AND $\mathcal{H}$ PART

In this section, we analyzed the influence of our $\mathcal{D}$ and $\mathcal{H}$ term for a single-branch MOptEq model trained without PE method shown in Table. 3. We set $\lambda = 0.02$ for the experiment. From the table, one can see that the contribution of the $\mathcal{H}$ term is slightly higher than the diversity term. But considering both two terms can significantly improve the results comparing with MOptEqs without considering these two terms.

---

[3]https://github.com/fastai/imagenette/

[4]https://www.image-net.org/

[5]https://s3.amazonaws.com/fast-ai-imageclas/imagenette2.tgz

| Model | Natural Accuracy |
|-------|-----------------|
| MOptEqs | **89.1**% |
| - $\mathcal{D}$ | 88.6% |
| - $\mathcal{H}$ | 88.4% |
| - $\mathcal{D}, \mathcal{H}$ | 87.4% |

Table 3: Evaluation on MOptEqs trained with and without considering $\mathcal{D}$ and $\mathcal{H}$ part.

## A.9    EXPERIMENTS ON THE IMAGENET

Except CIFAR and Imagenette, we also conducted experiments on ImageNet with 13M parameters shown as below:

| Model | Model Size | Natural Accuracy |
|-------|-----------|-----------------|
| ResNet | 13M | 70.3% |
| HR-Net | 13M | 72.3% |
| MOptEqs | 13M | **72.5**% |

Table 4: Evaluation on MOptEqs on ImageNet.

From the results, one can see that our model can achieve the satisfying performance on ImageNet compared with the state-of-the-art models.

## A.10    PE METHOD AND ROBUSTNESS

From the above experiments, one can see that the PE method may also improve the robustness of our MOptEqs. In this section, we conduct experiments to see whether the PE methods can further enhance the robustness of MOptEqs when we use adversarial training methods on these models. We finish comparisons of PGD-3 training for the single-view MOptEq with or without PE, and the result is shown as follows: From the table, we can

| Model | Natural Accuracy | PGD-20 Accuracy($\epsilon = \frac{8}{255}$) |
|-------|-----------------|-----------------|
| PGD-3 w/o PE | 78.60% | 37.18% |
| PGD-3 w. PE | 78.40% | **38.05**% |

Table 5: Evaluation on MOptEqs trained by PGD-3 with and without PE method.

conclude that our PE method can improve the robustness of our MOptEqs model trained by the adversarial training method. Such advantages also demonstrate the superiority of our models' mathematical interpretability since the PE method is acquired based on the model's hidden optimization problem.

## A.11    OTHER TRAINING METHODS: FIXED POINT METHOD AND ONE-STEP GRADIENTS

### A.11.1    METHODS

Since our model is not so complicated, we can use the fixed point method for $L$ iteration for the forward propagation. While for the back propagation, we can use BPTT (Mozer, 1989) and (Geng et al., 2021) proposed method, which only backward the final forward iterations using the chain rule:

$$\frac{\partial \mathcal{L}}{\partial (\cdot)} = \frac{\partial \mathcal{L}}{\partial \mathcal{T}_\Theta^L} \frac{\partial \mathcal{T}_\Theta^L}{\partial (\cdot)}$$

where $\theta \in \Theta$ are the learnable parameters for the implicit models, $\mathcal{T}_\Theta^L$ denotes the output $\mathbf{z}$ for the $L$-th fixed-point iteration. Since the back-propagation of such a strategy is similar

to backwards only a single layer of DNN module without the calculation of root-finding algorithm for the Jacobian in the implicit differentiation, the computational cost is much less than the implicit method. However, the gradient approximated by such a method is not accurate and may cause the performance drop. However, as we illustrated below, such a method can be regarded as a trade-off if the computational resources of training are not enough.

### A.11.2 EMPIRICAL RESULTS AND COMPUTATIONAL EFFICIENCY OF ONE-STPE GRADIENT

We take the ImageNette dataset as an example, the training memory cost, forward time, and test accuracy are shown as follows (each batch contains 32 images and is finished on 2 x GTX 1080Ti for 200 epochs):

| Model | Memory/Batch | Infer Time/Batch | Total Time | Accuracy |
|---|---|---|---|---|
| MDEQ | 8GB | 1.06s | 9h | 91.42% |
| MOptEq(one step) | 3GB | 0.26s | 3h | 91.21% |

Table 6: The comparison of the computational cost of MDEQ and our MOptEqs (trained by one-step gradient).

The results shows that the one-step gradient methods can save a lot computation resources while with slightly drop on the performance.

