# OpenReview forum: "Optimization inspired Multi-Branch Equilibrium Models"
_ICLR.cc/2022/Conference — ICLR 2022 Poster_

### Official Review · Reviewer_61kJ · 2021-11-02

**Correctness:** 3
**Technical Novelty And Significance:** 3
**Empirical Novelty And Significance:** 3
**Recommendation:** 6
**Confidence:** 1

**Main Review:**

Though authors include the concepts of interpretability in the motivation of this work. I cannot see the connections between interpretability to this work. Did authors define the interpretability as explanation of behaviors of CNN, such as LIME explains DNN for computer vision using the attribution to features? Could you please give an example to demonstrate how the proposal interpret the DNN.

**Summary Of The Paper:**

I don't quite understand this work. It seems authors proposed a new deep equilibrium model through incorporating a layer of optimization into the network. More specifically, authors may have extended the work MDEQ from two aspects: (1) inclusion of layers of deep neural network as optimizer and (2) introducing multi-branch architecture design. Experiments based on CIFAR demonstrate the advantage of the proposed solution in their settings. Theortical analysis with two propositions justify the properties of proposals.

**Summary Of The Review:**

I failed to collect evidence to reject this paper, so I believe it might be a solide work. I hope authors could address the interpretability issues here. I hope AC and SPC could hear from other reviewers.

---

> ### Author Response · Authors · 2021-11-20
> **Response to 61kJ**
>
> Thanks for your comments. We provide the following responses to your concerns.
>
> **1. About your concerns for model's interpretability**
>
> 1.  As we stated in various places in our paper, the “interpretability” we mainly discussed is “mathematical interpretability”. In contrast, LIME is so-called “Empirical interpretability” as discussed in the Related works—-Interpretability Section. LIME tries to use simple surrogate architectures like linear models to approximate the local behaviour of a DNN's black box. And then use the linear model as an explanation for DNNs. The surrogate models usually enjoy satisfying mathematical interpretability like SVM and linear model, which means we can mathematically understand why such models are designed. However, the approximation is usually unstable [1] and such methods can only tell us how the models perform instead of why it works. In order to know why it works, works have been to dissect DNNs intermediate features. But since implicit models like DEQs use accelerated algorithms as their inference procedure, such dissecting is not reasonable. For this count, the interpretability we discussed here is the "mathematical interpretability", whether the whole network structure can be summarized or explained as a mathematical system. For example, our MOptEqs will perform as our optimization problem defined. As shown in Figure2, the similarity matrix of two branches' output behaves similar to a diagonal matrix which means the diversities and branches of different branches can be preserved when using our design.
>
> [1] David Alvarez-Melis 1 Tommi S. Jaakkola. On the Robustness of Interpretability Methods. ICML-W, 2018.
>
> 2. "Mathematical interpretability" is essential for implicit models since such models' inference and the backward procedure is an implicit one by solving equations. For this account, dissecting its actual intermediate features is impossible. Then understanding the implicit models from its behind mathematical systems are important. For example, Neural ODEs can be analyzed from dynamic systems, and many interesting works have been proposed inspired by various dynamic systems. From such a perspective, MDEQs and most DEQ models perform almost a black-box. However, our model can be analyzed through our model's hidden objective function. All the components of our MOptEqs have a clear purpose defined in the hidden optimization problem.
>
>
> 3. Such interpretability can be used to design implicit models. Like new Neural ODEs can be designed from the inspirations from various dynamic systems. Our HH&D structure is obtained from the perspective of constructing an optimization problem first and then using its first-order condition as our MOptEqs equilibrium equations. Furthermore, our PE training method is also obtained by our analyzing our model's hidden optimization problem since the reconstructed perturbation is to perturb the objective value of the hidden objective function as large as possible. As we have shown in the experiments, such perturbation and strategy are useful.
>
> 4. The mathematical reasons behind our MOptEqs are minimizing least-square problems with regularizers to constrain the features. Although our fusion structure is a little complicated and not self-evident, we can explain its purpose by the hidden objective functions.

---

### Official Review · Reviewer_DkRL · 2021-11-02

**Correctness:** 4
**Technical Novelty And Significance:** 2
**Empirical Novelty And Significance:** 2
**Recommendation:** 6
**Confidence:** 2

**Main Review:**

Strengths:
1. The proposed method is straight forward and easy to follow, the newly proposed terms in the objective function have good motivations;
2. Model sizes in multiple-scale experiments are much smaller than related works, which illustrates the effectiveness and efficiency of the proposed method;

Weaknesses, questions & comments:
1. The performance improvements seem to be trivial, although it is impressive to see the model size comparison in multiple-scale experiment. It would be great if the author could provide comparisons of different methods with same model size (results of MDEQ with smaller model size and results of MOptEqs with larger model size). It would be interesting to see whether the proposed method also has good scalability;
2. More experiments on higher dimensional datasets are suggested;
3. Could the author provide some ablation studies investigating the contribution of each component (inner-product term and diversity term)?
4. Are the compared methods in tables trained with any kind of perturbation or not? It would clearer if all the methods are compared in two different settings: with and without perturbation;

**Summary Of The Paper:**

The paper propose a new method which can efficiently utilize inputs with different scales, the propose method obtains better quantitative results in experiments.

**Summary Of The Review:**

This paper propose a new method, which can handle different input scales. The  proposed method is well-presented with good motivation, obtains better results with smaller model sizes in experiments. There are several questions remaining, I'm looking forward to the responses and discussions.

---

> ### Author Response · Authors · 2021-11-20
> **Response to Reviewer DkRL**
>
> Thanks for your comments. We provide the following responses to your concerns.
>
> ---
> ---
>
> **1. About your concerns for the model's scalability and requests for more expeirment results**
>
> We've finished running experiments for a larger MOPtEq model with around 10M parameters on CIFAR-10, CIFAR-100 and ImageNette. And we listed the results for small models ($\approx$2M) and large models ($\approx$ 10M) in the following:
>
> The classification accuracy for CIFAR-10:
>
> |   Model |   SMALL  | LARGE |
> |  :----:  | :----:  | :----:  |
> | MDEQ  | $92.6$% | $93.8$% |
> | MOptEq  | $94.0$% | $94.6$% |
>
> The classification accuracy for CIFAR-100:
>
> |   Model |   SMALL | LARGE |
> |  :----:  | :----:  | :----:  |
> | MDEQ  | $70.8$% | $72.4$% |
> | MOptEq  | $73.4$% | $74.7$% |
>
> For ImageNette:
>
> |   Model |   SMALL  | LARGE |
> |  :----:  | :----:  | :----:  |
> | MOptEq  | $90.5$% | $91.3$% |
> | MDEQ  | $92.1$% | $92.4$% |
>
>
> Furthermore, we also finished a 13M model for ImageNet shown as below:
>
> |   Model |   Model Size  | Accuracy |
> |  :----:  | :----:  | :----:  |
> | ResNet18 | 13M | $70.2$% |
> | HRNet  | 13M | $72.3$% |
> | MOptEq  | 13M | $72.5$% |
>
> From the results, one can see that the scalability of our models is satisfying. It can achieve satisfying results compared with the widely used models of the same size.
> All experimental results have been updated in the new version.
>
>
> ---
> ---
>
>
> **2. About your suggestions on the contribution of each component:**
>
> We've finished experiments for this ablation study as shown below:
>
> |   Model |   Accuracy |
> |  :----: | :----:  |
> | MOptEq  | $\mathbf{89.1}$% |
> | -$\mathcal{D\phantom{,H}}$  |  $88.6$%  |
> | -$\mathcal{H\phantom{,H}}$  |  $88.4$%  |
> | -$\mathcal{H,D}$  |  $87.4$%  |
>
> From the results, one can see that the contributions of the inner-product are slightly higher than the diversity term. Both of them shows a large improvement compared with MOptEqs without considering these two terms.MOptEqs without adding
>
> ---
> ---
>
>
> **3. About your suggestions on testing other models with perturbations:**
>
> We need to clarify that the reconstructed perturbations can only be implemented in our MOptEq models. For this count, we only conducted experiments for MDEQ with Gaussian noise of the same size as our MOptEqs for MDEQ's input features. However, the difference is negligible. We think such a phenomenon is reasonable since such simple noise for features will not influence the results much, as shown in our ablation studies Figure.3(a).
>
> ---
> ---

---

> > ### Comment · Reviewer_DkRL · 2021-12-01
> > **My major concerns have been addressed**
> >
> > I've read the authors' response. My major concerns have been addressed, thus I will raise the score.

---

### Official Review · Reviewer_8Er5 · 2021-11-02

**Correctness:** 3
**Technical Novelty And Significance:** 2
**Empirical Novelty And Significance:** 2
**Recommendation:** 5
**Confidence:** 3

**Main Review:**

Pros:
* Even though incorporating different input resolutions to improve the performance of the model is not novel, I believe that the design of the architecture to incorporate multiple scales is still an interesting problem.
* I checked the proofs and I believe that the work is technically correct.
* Empirically, the proposed method achieves impressive results on CIFAR-10, CIFAR-100, and ImagImagenette and it is convincing that the proposed method outperforms MDEQ on these small-scale experiments.

Concerns:
* I have one concern about the additional computational/memory overhead of “unrolling”. I believe that the model size (number of parameters) is relatively small for CIFAR-10 experiments. However, for larger models (and more complex datasets such as ImageNet), do you think your method would scale to these settings?
* Following up on the previous concerns, I recommend the authors clearly state the additional overhead (wall-clock time/additional memory) in the results section to describe the potential limitations of the approach.
* Another concern I have is that, since the proposed method is trained by unrolling, I wonder if it is fair to compare against MDEQ? I believe that the direct comparison is unfair. For example, the proposed method does not have useful properties of equilibrium models (e.g. constant memory). However, we are solely comparing the performance between MDEQ and the proposed approach.
* In the introduction, the authors mention that OptEqs have “mathematical interpretability”. Does this necessary correspond to, is it easier to interpret what the network is doing? If not, what is the core benefit of “mathematical interpretability”?

**Summary Of The Paper:**

The paper introduces a multi-scale network (called Multi-Branch OptEqs or MOptEqs) that extends on recent results of [1]. Specifically, the authors propose to use several DEQ models to incorporate samples of multi-resolutions. The authors further propose a modified training strategy (called perturbation enhanced training) to improve the performance of the model. Empirically, the proposed model shows better performances compared to the previous DEQ models on CIFAR-10, CIFAR-100, and Imagenette datasets.

[1] Xie, Xingyu, et al. "Optimization Induced Equilibrium Networks." arXiv preprint arXiv:2105.13228 (2021).

**Summary Of The Review:**

I believe the idea of the paper is interesting and the authors show the impressive results on CIFAR experiments. However, there are some concerns regarding scalability (since the proposed method performs unrolling), additional overhead, and experimental fairness. Most importantly, the proposed approach is trained with unrolling to a fixed depth and does not have a property of equilibrium models. Hence, I give a score of 5.

---

> ### Author Response · Authors · 2021-11-20
> **Response to Reviewer 8Er5:**
>
> Thanks for your comments. We provide the following responses to your concerns.
>
> ---
> ---
>
> **1. About your worries for unrolling:**
>
> We would like to clarify that we do not use "unrolling" for the ICLR version. We forward our MOptEqs and obtain its backward gradient using the same method as MDEQ and other DEQ models as we stated in Section.2.3. We use accelerated solvers for our MOptEqs forward and the same implicit differentiation method to approximate the gradients. For this count, worries about the unfairness brought by the "unrolling" are unnecessary.
>
> ---
> ---
>
>
> **2. About the memory cost:**
>
> Since we use the same method for inference and backward, the peak memory cost for MDEQ and our MOptEqs are almost the same. Take ImageNette with $256\times 256$ input size for an example, the memory usage for MDEQ and MOptEqs with batch_size=32 are shown below:
>
> |   Model |   Model Size  | Peak Memory | Training Time for 1 Batch |
> |  :----:  | :----:  | :----:  | :----:  |
> | MDEQ  | 10M | 7.5GB | 2.5s  |
> | MOptEq  | 10M | 7GB |  2.3s |
>
> As one can see, the memory cost and training speed are slightly better than MDEQ. Such difference is caused by the structure difference and hyper-parameters for the accelerated solvers since we use the same training method.
>
> ---
> ---
>
>
> **3. For Large Models and other datasets.**
>
> We've finished running experiments for a larger MOPtEq model with around 10M parameters on CIFAR-10, CIFAR-100 and ImageNette. The results are listed below:
>
> For CIFAR-10:
>
> |   Model |   Model Size  | Accuracy |
> |  :----:  | :----:  | :----:  |
> | MDEQ  | 10M | $93.8$% |
> | MOptEq  | 1.9M | $94.0$% |
> | MOptEq  | 8.1M | $94.6$% |
>
> For CIFAR-100:
>
> |   Model |   Model Size  | Accuracy |
> |  :----:  | :----:  | :----:  |
> | MDEQ  | 10M | $72.4$% |
> | MOptEq  | 1.9M | $73.4$% |
> | MOptEq  | 8.1M | $74.7$% |
>
> For ImageNette:
>
> |   Model |   Model Size  | Accuracy |
> |  :----:  | :----:  | :----:  |
> | MDEQ  | 10M | $91.3$% |
> | MOptEq  | 1.9M | $92.1$% |
> | MOptEq  | 10M | $92.4$% |
>
>
> Furthermore, we also finished a 13M model for ImageNet shown as below:
>
> |   Model |   Model Size  | Accuracy |
> |  :----:  | :----:  | :----:  |
> | ResNet18 | 13M | $70.2$% |
> | HRNet  | 13M | $72.3$% |
> | MOptEq  | 13M | $72.5$% |
>
> From the results, one can see that the scalability of our models is satisfying. It can achieve satisfying results compared with the widely used models of the same size.
> All experimental results have been updated in the new version.
>
>
> ---
> ---
>
> **4. About the necessities for the mathematical interpretability:**
>
> The mathematical interpretability is necessary for the following reasons:
>
> 1. Mathematical interpretability can also be used to understand how the model works. For example, the mathematical reasons behind our MOptEqs are minimizing least-square problems with regularizers to constrain the features. Although our fusion structure is a little complicated and not self-evident, we can explain its purpose by the hidden objective functions.
>
> 2. It provides us with a new way for DEQs block construction. Although the construction of explicit models has been experimentally or theoretically explored for years and has raised many optimal architectures, no evidence shows that the optimal blocks for explicit models can also perform the best. For the implicit models with well mathematical interpretability like Neural ODEs, they proposed their well-performed structure from the motivation of the analysis on the dynamic systems. Our models can also be modified for different tasks by modifying the model's hidden objective functions instead of repeating the block searches as researchers been done for explicit models.
>
> 3. Furthermore, the implicit models performs more likely to be a black box compared with DNNs due to their implicit inference and backward procedure. Dissecting the black-box is much harder than DNNs, which makes the "mathematical interpretability " much important for implicit models.
>
>
> 4. Former machine learning tasks can be interpreted by mathematical problems like dictionary learning, SVM, etc. These methods and traditional optimization algorithms are robust in many practical tasks compared with deep models, proving the importance of the deep model’s mathematical interpretability.

---

> ### Author Response · Authors · 2021-12-02
> **Need further clarification?**
>
> Thanks for your comments. We have tried our best to address the concerns. Since our work only proposes a new equilibrium model, we can choose the root-finding algorithm or other methods for the forward and backward propagation. But the choice of the algorithms may influence the results as your concern. For this account, we use the same scheme and algorithm for inference and backward as MDEQ's repo does in order to make the comparison fair. Could you please re-evaluate our work?

---

### Official Review · Reviewer_2WhH · 2021-11-08

**Correctness:** 4
**Technical Novelty And Significance:** 3
**Empirical Novelty And Significance:** 3
**Recommendation:** 6
**Confidence:** 3

**Main Review:**


Strengths:
1. Performance: The proposed method obtains decent improvement in performance and reduction in model size as compared to prior work on this topic.
2. Explainibility and ablation studies: The detailed ablation study shows the effect of design  choices such as Hierarchical Heritage and Diversity modules, perturbation size and regularizers.


Weaknesses/Questions
1. Writing: The paper is not well-written and organized. The key ideas are not properly organized and explained. For someone who is not familiar with the implicit models literature, I had to spend  time carefully re-reading the paper to make sure that I understood all the details.
2. Somewhat limited novelty?: The key ideas proposed in the paper (multi-scale, Hierarchical Heritage, Diversity) are quite common in the explicit models (i.e., standard neural networks). It is not clear to me if adapting them directly to implicit models is quite a significant advance.
3. How would the approach scale and compare to state-of-the-art architectures (or even a larger ResNet) trained on ImageNet?


**Summary Of The Paper:**

This paper proposes a new implicit model for deep learning called Multi-branch Optimization induced Equilibrium networks (MOptEqs), which models the hidden objective function for making efficiently use of different scales inputs. The fusion model uses two ideas:  Hierarchical Heritage, which ensures the similarity of corresponding channels of the near branches and (2)Diversity Module, which optimizes diversity in non-corresponding channels of neighboring braches. Taken together, these design choices lead to better performance and smaller model size on image recognition tasks on CIFAR-10 and CIFAR-100 datasets, as compared to  prior work on Equilibrium networks that use single/multiple branch architectures.


**Summary Of The Review:**

The paper proposes some interesting ideas for using multi-scale inputs and feature hierarchy and diversity for training equilibrium models. The results on CIFAR-10 look promising, but I have some concerns about the writing and the novelty of the work.

---

> ### Author Response · Authors · 2021-11-20
> **Response to Reviewer 2WhH:**
>
> Thanks for your comments. We provide the following responses to your concerns.
>
> ---
> ---
>
> **1. About Writings:**
>
> We have entirely rewritten the front part of our paper for readers who are unfamiliar with DEQ models. They are mainly listed below:
>
> 1.  We rewrite the introduction of OptEqs to ensure readers know how the model inference. We think readers who are not familiar with DEQ may become confused at this point in the first version. The forward propagation of OptEqs and other DEQ models are all trying to solve the equilibrium equation. However, the performance for OptEqs is still weak.
>
> 2. Secondly, we rewrite the introduction parts for MDEQ. Due to its forward propagation being implemented by solving equations, MDEQ and other DEQs are entirely a black box that lacks interpretability. Furthermore, no evidence shows that the optimal neural block architecture can perform best in the implicit scheme. So the new blocks for DEQs are still worth exploring. These reasons motivated our structure.
>
> 3. Rewrite the DEQ parts in the Implicit Models. The central part of DEQ models is their equilibrium equations. Moreover, its construction is still worth exploring.
>
> 4. Rewrite the first paragraph in Section2.1 try to make readers know how our MOptEqs is designed. Since our MOptEqs' equilibrium equations are just the first-order conditions for MOptEqs' hidden optimization problem, we first try to model our model's objective functions by analyzing the tasks. Then we propose two regularizers for our optimization problem, and the two terms behave as fusion modules when we translate them to neural blocks by calculating the first-order conditions for the hidden objective function.
>
>
> ---
> ---
>
> **2 About the novelty:**
>
> Although the two modules are inspired by the findings in the explicit models, their designs are different.  Our modules are designed by reformulating the hidden objective function of the MOptEqs. For this count, the performance of our module can be mathematically ensured and explained. Moreover, as we can see from Figure 1(b), the propagation of our new modules is not trivial. Like SIFT features compared with deep neural features, SIFT is controllable since it can be mathematically ensured, while deep neural features can behave unreliably since different data may lead to different types of features. The novelty of our two modules is not attributed to their motivation or purpose. We think the contribution of our two modules is that it proves that we can obtain some new structure by reformulating the hidden objective function for the tasks, and such structure can indeed achieve better performance. Moreover, as illustrated in the ablation studies, our new modules can achieve much better results than simply considering these terms in the Loss functions.
>
> Besides, we proposed a novel PE strategy for our MOptEq due to our analysis of the hidden optimization problem. The method can not only enhance its natural performance but can also make it more stable.
>
> As for the overall impact, we think it inspires us to design our neural architectures from the aspect of designing the optimization function.
>
> ---
> ---
>
> **3 About scaling to large models and ImageNet:**
>
> We've finished running experiments for a larger MOPtEq model with around 10M parameters on CIFAR-10, CIFAR-100 and ImageNette. The results are listed below:
>
> For CIFAR-10:
>
> |   Model |   Model Size  | Accuracy |
> |  :----:  | :----:  | :----:  |
> | MDEQ  | 10M | $93.8$% |
> | MOptEq  | 1.9M | $94.0$%|
> | MOptEq  | 8.1M | $94.6$% |
>
> For CIFAR-100:
>
> |   Model |   Model Size  | Accuracy |
> |  :----:  | :----:  | :----:  |
> | MDEQ  | 10M | $72.4$% |
> | MOptEq  | 1.9M | $73.4$% |
> | MOptEq  | 8.1M | $74.7$% |
>
> For ImageNette:
>
> |   Model |   Model Size  | Accuracy |
> |  :----:  | :----:  | :----:  |
> | MDEQ  | 10M | $91.3$% |
> | MOptEq  | 1.9M | $92.1$% |
> | MOptEq  | 10M | $92.4$% |
>
>
> Furthermore, we also finished a 13M model for ImageNet shown as below:
>
> |   Model |   Model Size  | Accuracy |
> |  :----:  | :----:  | :----:  |
> | ResNet18 | 13M | $70.2$% |
> | HRNet  | 13M | $72.3$% |
> | MOptEq  | 13M | $72.5$% |
>
> From the results, one can see that the scalability of our models is satisfying. It can achieve satisfying results compared with the widely used models of the same size.
> All experimental results have been updated in the new version.
>
> ---
> ---

---

> > ### Comment · Reviewer_2WhH · 2021-12-01
> > **Response to rebuttal**
> >
> > Thank you for updating the text and the additional experiments. I am satisfied with the response and recommend acceptance of the paper.

---

### Author Response · Authors · 2021-11-21
**A Summary of Paper Updates**

Thanks for all the comments. We update some parts in our submission:

1.  We rewrite the front part of our submission to make the readers know how equilibrium models work more clearly. They use accelerated algorithms to solve its central part: the equilibrium equation and the solution of the equilibrium equation is the equilibrium models' output. Due to the implicit procedure for their inference, most DEQ's are entirely black boxes. Moreover, how to construct the efficient DEQ structure is also an open problem and have not been fully explored. For this count, we proposed our MOptEqs architecture with good performance and satisfying mathematical interpretability since the output of our model is the first-order stationary points of a hidden objective function we designed.

2. We've updated some new empirical results for large models on CIFAR, ImageNette and ImageNet.

---

### Decision · Program_Chairs · 2022-01-20

**Decision:**

Accept (Poster)

**Comment:**

The paper proposes a multi-scale network that uses DEQ models to incorporate samples at multiple resolutions. The authors also propose a training strategy to improve the performance of the model. The authors investigate the interest of the approach through ablation and explainability, weighing the value of hierarchical heritage, diversity modules, perturbation size, and regularization penalties.

The reviewers appreciated that the authors tackled the problem of incorporating multiple scales and the “impressive results” on CIFAR-10, CIFAR-100. The reviewers also expressed concerns regarding the computational assessment, in particular the additional computational/memory overhead of unrolling and what the authors mean by “explainability” in their experimental evaluation. The reviewers also made suggestions to organize the paper better.

The authors submitted responses to the reviewers' comments. After reading the response, updating the reviews, and discussion, the reviewers who took part in the discussion considered that they are “satisfied by the response” and the “major concerns have been addressed”.  The feedback provided was already fruitful and the final version should be already improved. The ablative analysis and comparison to baseline is careful and thorough.

Accept. Poster.